# Exploiting TCR Repertoire Analysis to Select Therapeutic TCRs for Cancer Immunotherapy

**DOI:** 10.3390/cells14151223

**Published:** 2025-08-07

**Authors:** Ursule M. Demaël, Thunchanok Rirkkrai, Fatma Zehra Okus, Andreas Tiffeau-Mayer, Hans J. Stauss

**Affiliations:** Institute of Immunity and Transplantation, Division of Infection and Immunity, University College London, Royal Free Hospital, London NW3 2PP, UK; thunchanok.rirkkrai.21@ucl.ac.uk (T.R.); fatma.okus.20@ucl.ac.uk (F.Z.O.); andreas.mayer@ucl.ac.uk (A.T.-M.)

**Keywords:** T cell receptor, TCR sequencing, TCR repertoire, tumour infiltrating lymphocytes, meta-clonotype, cancer immunotherapy, TCR-T cells, clonality, cell therapy

## Abstract

Over the past decade, numerous innovative immunotherapy strategies have transformed the treatment of cancer and improved the survival of patients unresponsive to conventional chemotherapy and radiation therapy. Immune checkpoint inhibition approaches aim to block negative regulatory pathways that limit the function of endogenous T cells, while adoptive cell therapy produces therapeutic T cells with high functionality and defined cancer specificity. While CAR engineering successfully targets cancer surface antigens, TCR engineering enables targeting of the entire cancer proteome, including mutated neo-antigens. To date, TCR engineering strategies have focused on the identification of target cancer antigens recognised by well-characterised therapeutic TCRs. In this review, we explore whether antigen-focused approaches could be complemented by TCR-focused approaches, whereby information of the TCR repertoire of individual patients provides the basis for selecting TCRs to engineer autologous T cells for adoptive cell therapy. We discuss how TCR clonality profiles, distribution in T cell subsets, and bioinformatic screening against continuously improving TCR databases can guide the selection of TCRs for therapeutic application. We further outline in vitro approaches to prioritise TCR candidates to confirm cancer reactivity and exclude recognition of healthy autologous cells, which could provide validation for their therapeutic use even when the target antigen remains unknown.

## 1. Introduction

The immune system plays a critical role in cancer surveillance, engaging a range of innate and adaptive immune cells, including B cells, natural killer (NK), natural killer T cells (NKT), and γ/δ T cells. Over the past few decades, the pivotal role played by endogenous *αβ* T cells in anti-tumoural immunity has been a particular focus of study and opened up entirely new treatment modalities in oncology.

Early pre-clinical insights into the significance of the immune system in tumour control included observations that mice with immune defects are more susceptible to spontaneous and carcinogen-induced cancers and clinical observations of increased cancer incidence in immunodeficient patients, in settings like AIDS or post-organ-transplant immunosuppression [1]. In the late 1980s, proof of concept that harnessing the immune system can provide anti-tumoural therapeutic benefits came with seminal trials including re-infusion of ex vivo expanded tumour-infiltrating lymphocytes (TILs), which mediated regression in patients with metastatic melanoma [2]. Early trials in the 1980s also showed that delivering the T cell stimulatory cytokine IL-2 can lead to tumour regression in melanoma and renal cell carcinoma (RCC) [3], while landmark clinical trials in the early 2010s established that monoclonal antibodies (mAbs) blocking the T cell inhibitory checkpoint molecules CTLA-4 and PD-1 led to improved survival in several cancer types [4,5].

Collectively, these studies all pointed to the presence of an endogenous pool of T cells that specifically recognises cancer cells and, when amplified, boosted, or released from inhibition, can potently eliminate tumours. Accordingly, a dominant, although simplistic, paradigm has been to classify tumours as either “hot” or “cold” based on the degree of immune activity [6], with “hot” T cell-infiltrated tumours, including melanoma, bladder cancer, and non-small cell lung cancer (NSCLC) exhibiting favourable responses to immune mobilising therapies, while “cold” tumours without infiltration of functionally competent immune cells such as glioblastoma (GBM), pancreatic cancer, or prostate cancer being refractory [7]. The classification of tumours into “hot” or “cold” based on their immune infiltration predicts an improved prognosis for cancer patients with high levels of pre-treatment T cell infiltration and a favourable subset composition and T cell phenotype [6,8].

In this review, we will explore the working hypothesis that the TCR repertoire of tumour-infiltrating T cells contains information that is relevant for the prognosis of cancer patients, and, more importantly, that it is possible to select certain TCRs for cancer therapy with engineered autologous T cells even when the peptide specificity of identified TCRs remains unknown.

## 2. T Cells in Cancer Immunity

In the last two decades, treatment options in oncology have been truly revolutionised by entirely new classes of immunotherapies, many of which can be conceptualised as pro-drugs which all converge to the final active drug product of functionally competent anti-tumour T cells (Figure 1). Adoptive cell therapies include ex vivo expanded TIL infusion, which has recently received FDA approval for unresectable or metastatic melanoma, and transfer of T cells engineered with receptors that redirect the specificity of T cells to cancer antigens.

The dominant modality has been Chimeric Antigen Receptor T cells (CAR-T cells), primarily directed against CD19 and BCMA B cell lineage antigens, whose impressive clinical remissions in haematological malignancies have resulted in seven regulatory approvals, although similar success has not yet been reached with solid tumours. CARs are synthetic receptors composed of an antigen-binding domain derived from antibody single-chain variable fragments (scFvs), which restricts their recognition to cell surface targets. These binding domains are fused to intracellular signalling domains, typically including CD3ζ and one or more co-stimulatory molecules such as CD28 and 4-1BB. In contrast, T cell receptor (TCR)-engineered T cells can recognise cell surface and intracellular antigens presented by Major Histocompatibility Complex (MHC) molecules, offering broader antigen coverage, including the intracellular onco-proteome and neoantigens, thus addressing some of the limitations of CAR-T cell therapies. TCR-T cells have gained significant attention, marked by the recent FDA approval of afami-cel, a TCR-T therapy targeting MAGE-A4 in patients with metastatic synovial sarcoma [9].

### 2.1. TCRs Govern T Cell Function

The specificity of T cells is underpinned by their surface TCRs, which mediate antigen recognition by binding to short peptides presented on MHC molecules, referred to as Human Leukocyte Antigen (HLA) in humans. A hugely diverse and clonally distributed TCR repertoire is generated in the thymus by random recombination of TCR*α*/*β* gene segments, with junctional diversification, followed by positive and negative selection, producing a highly diverse TCR repertoire that is tolerant to self antigens and capable of recognising non-self antigens.

Structurally, the TCR is a heterodimeric receptor composed of a disulfide-linked *α* and *β* chain, further associated non-covalently with CD3εγδζ chains, which are required for TCR expression on the cell surface and for intracellular signal transduction. Through its hypervariable complementarity-determining regions (CDRs), the TCR*αβ* expressed by CD8+ and CD4+ T cells interacts with short peptides presented on MHC-class I (*HLA-A,B,C*) and MHC-class II (*HLA-DR, DP, DQ*), respectively. Whilst MHC class I presented peptides are derived primarily from the pool of nuclear, cytosolic, and surface proteins expressed by a cell, MHC class II peptides are mostly derived from proteins that are present in the extracellular compartment. Thus, MHC class I-restricted CD8+ T cells can recognise peptides sampling the entire proteome of the target cell, whilst MHC class II-restricted CD4+ T cells primarily recognise peptides of proteins that are present in the microenvironment, including proteins released from dying tumour cells.

### 2.2. Cancer Targets Recognised by TCRs

TCRs recognise a broad range of tumour antigen classes, including viral antigens from oncogenic viruses such as EBV, HBV, and HPV, neoantigens generated by tumour-specific mutations, and tumour-associated antigens (TAA) such as cancer testis antigens (CTA) and tissue-specific differentiation antigens, whose relative advantages and limitations as therapeutic targets have been extensively reviewed elsewhere [10,11]. In short, there is a limited pool of therapeutically tractable antigen targets fulfilling key criteria such as immunogenicity, tumour specificity, and oncogenic relevance [12]. Beyond canonical protein-coding sequences, a growing body of literature has explored contributions from the “dark proteome”, including peptides arising from endogenous retroviruses and transposable elements [13], aberrant splice variants [14], frameshift mutations [14], circular RNAs [15], and microbial sources [16,17], all of which are processed intracellularly and presented on the cell surface via MHC, together expanding the potential repertoire of therapeutically actionable TCR targets.

A major bottleneck in TCR immunotherapy development is the difficulty of specifically identifying tumour-reactive TCRs, especially since the vast majority of TCRs remain “orphan”, with unknown antigen specificity. As a result, there is a growing interest in shifting from antigen-centric discovery to TCR-centric frameworks, focusing on analysing the TCR composition, diversity, and clonal dynamics in the tumour microenvironment (TME) as proxies for immunogenic activity and therapeutic exploitation, even when the peptide-specificity of the selected TCRs remains unknown.

## 3. Insights from TCR Composition and Diversity in the TME

As discussed above, the abundance and composition of TILs are prognostic indicators for patients with cancer. Quantitative assessments using digital pathology provide consistent evidence that CD4+ and CD8+ abundance are associated with better prognosis [18], while integration with gene-expression profiling and analysis of additional cell types including B cells, NK cells, and myeloid cells [19] further refine how specific T cell phenotypic signatures and broader cellular context interact to favour protective anti-tumoural responses. An important tool to understand drivers of protective vs. non-protective T cell responses in tumours is analysis of TCR diversity and clonal expansion. In addition to its mechanistic role in antigen recognition, the TCR serves as a unique molecular barcode, enabling precise spatial and temporal tracking of individual T cell clones and inference of T cell activity in tumours.

### 3.1. Metrics of TCR Diversity and Clonal Distribution

Important metrics to analyse the composition and distribution of TCR repertoire are indices of diversity and clonality, whose mathematical definitions come from the field of ecology and information theory.

Repertoire *richness* reflects the number of unique TCR sequences, while repertoire *evenness* encompasses inequalities in the clonal abundances of TCR sequences. Commonly used metrics of TCR diversity such as Shannon and Simpson diversity [20] account for both *richness* and *evenness* (Figure 2), thus assigning a lower diversity to a repertoire marked by uneven oligoclonal expansions [21]. Simpson diversity is more sensitive to clonal dominance than other measures of diversity and thus is a particularly useful metric for detecting potential oligoclonal expansions within tumour-infiltrating repertoires.

An important caveat for measuring metrics of diversity is that they can be confounded by variation in sampling depth, namely the total number of TCR sequences captured and sequenced per sample [22]. Low-frequency clonotypes can be missed when sampling depth is insufficient, leading to underestimation of diversity. Computational subsampling of repertoires to equal sizes provides a means to address this problem, as does the use of metrics such as Simpson’s diversity for which unbiased estimators exist [23,24].

Lastly, *overlap* indices such as the Jaccard index can measure the similarity between TCR repertoires, which is useful for comparing repertoires before and after treatment, comparing TCR repertoires across tissue compartments or even across patients. Indices that are sensitive to variation in the most clonally expanded TCRs might again be particularly well suited to capture immunological events driving dominant anti-tumoural responses.

Despite a suite of metrics to quantify diversity, there is no robust consensus in the field about the prognostic value of TCR diversity in tumours [25,26,27]. While high TCR diversity potentially reflects a broad pool of TCR clones able to recognise tumour antigens, clonal dominance of certain TCRs may reflect expansion of tumour-reactive T cells. Additionally, clonotypes may be present at low levels of expansion if they are directed against a sub-dominant tumour antigen but might still be useful targets for immunotherapy.

### 3.2. Predictive Power of TCR Clonality Pre and Post Treatment

In a cohort of almost 200 patients with primary melanoma, the baseline Simpson clonality of intra-tumoural TCR*β* chains could not differentiate progressors versus non-progressors, when correcting for variables such as tumour thickness and patient age [28]. Likewise, the overall TCR richness and evenness of the baseline tumour did not differ between responders and non-responders to TIL therapy in the context of metastatic melanoma [29]. Instead of global metrics of diversity, the most informative variables in this study were the number of CD8+ clones at baseline with experimentally confirmed tumour reactivity, and the number of these dominant clones carried over in the TIL product for re-infusion, yielding a TIL product with a higher absolute number of tumour-reactive clones.

The change in T cell composition and diversity pre- and post-immunotherapy has also been assessed in several case reports and observational studies. PD-1 blockade increased intra-tumoural proliferating (Ki67+) CD8+ T cells in melanoma [7], with responders additionally displaying a lower overall richness and a greater clonality, namely a higher number of expanded clones, after treatment, with similar results described in peripheral blood in the context of lung cancer [30]. The clonal repertoire of exhausted CD8+ T cells was largely replaced by novel clones after PD-1 therapy in patients with basal cell carcinoma (BCC) or squamous cell carcinoma (SCC), suggesting the importance of mobilising new, highly clonally expanded specificities that were not present in the pre-treatment tumour [31].

### 3.3. TCR Sharing Between Blood and Tumour

Importantly, recent studies have shown rates of substantial discordance between TCR clonal frequencies in peripheral blood and tumour tissue [32,33,34]. The implications are firstly that TCR sequences in PBMCs are not necessarily reliable proxies of immunological events in the TME but secondly open up a strategy to infer tumour-reactive clones by detecting TCR clones differentially expanded in the tumour compared to the periphery. Clonal expansion is a useful, although imperfect, proxy of proliferation to tumour antigen, as it can reflect responses to peptides present in the local tissue environment or antigen-nonspecific expansion by cytokines, and should thereby be performed where possible with controls of adjacent non-tumour tissue [35].

### 3.4. Public TCR Responses Shared Across Patients

The majority of TIL-derived TCRs remain highly private, given the stochastic nature of recombination and the individual landscape of somatic mutations and HLA types, posing a significant challenge for therapeutic translation, as most TCRs are unique to individual patients.

Despite the low *a priori* probability of two individuals sharing identical TCR sequences generated by random recombination events, there are consistent reports of “public” TCRs or TCR motifs shared between individuals in several cancer types [36,37]. In a comparative analysis of EBV-driven nasopharyngeal carcinoma (NPC) and non-viral head and neck squamous cell carcinoma (NSCC), Teng et al. identified a small fraction of TCR*β* clonal overlap, reporting that both cancer types contained “public” clonotypes (i.e., clonotypes shared between patients) but that these were unique to that cancer type [38]. Together, these data suggest that public TCRs might arise due to shared, cancer-specific antigenic footprints and hint at the promise of using public TCRs as probes to detect common antigen drivers, likely to be public TAAs rather than private cancer neo-antigens.

### 3.5. TCR Meta-Clonotypes

The concept of meta-clonotypes was recently introduced to group structurally similar TCRs that are inferred to share antigen specificity [39] (Figure 3). Unlike traditional clonotypes, which are defined by identical nucleotide sequences, meta-clonotypes cluster TCRs with highly similar CDR3 sequences, shared motifs, or comparable biochemical features. Biochemical features refer to the biochemical properties of amino acids at a given CDR3 position, rather than amino acid identity alone, such as hydrophobicity [40]. This approach facilitates the study of “public” T cell responses by overcoming the low likelihood of observing exact sequence matches.

Despite having highly similar TCR sequences, members of a given TCR meta-clonotype family can have divergent phenotypic profiles and fates in the tumour. For example, in a longitudinal study of a patient with NSCLC receiving anti-CTLA-4 and anti-PD/PDL-1, single-cell RNA sequencing of meta-clonotypes identified by highly similar TCR*β* chains revealed that they originated from phenotypically different clusters and displayed different expansion and contraction dynamics over the course of immune checkpoint blockade therapy [41].

Importantly, immune responses recruiting an ensemble of closely related TCRs within a meta-clonotype are thought to confer more robust and durable protection. This functional redundancy not only buffers against antigenic escape through epitope mutation but also provides resilience in the face of T cell dysfunction or clonal attrition. We predict that members of a TCR meta-clonotype displaying reactivity to a given tumour antigen will display a range of TCR affinities which can impact the kinetics [42] as well as the nature [43] of T cell responses.

As such, quantifying the breadth and clonal architecture within meta-clonotypes holds promise as a biomarker of effective anti-tumour immunity and may inform the rational design of TCR-based diagnostics and therapeutics.

### 3.6. Which T Cell Subsets Harbour Clonal Expansion?

The phenotypes of T cells harbouring clonally expanded TCRs span distinct subsets, most commonly antigen-experienced populations exhibiting an effector memory or exhausted phenotype, characterised by markers including LAG3, TIGIT, PDCD1, TIM-3 and CTLA-4. This association underscores the fact that the phenotype and functional state of T cells provides crucial context for interpreting clonal expansion and identifying tumour-specific responses.

In a study integrating over 23,000 TCR clonotypes from several metastatic cancer types, Lowery et al. identified that the majority of TCR clones were singletons, unique TCRs represented by a single copy in a sample, (83%), and that oligoclonal expansion was concentrated in dysfunctional, resident memory and effector memory clusters, which also preferentially contained the tumour-reactive TCRs [44]. Likewise, the most expanded clones in anti-PD-1-treated basal and squamous cell carcinoma [31] were found within exhausted CD8+ T cells, highlighting that strong tumour-specific expansion may also drive exhaustion.

The enrichment of clonally expanded TCRs in exhausted phenotypes poses an experimental challenge, as these TIL may be too exhausted to enable ex vivo screening, providing a rationale for TCR screening approaches “rescuing” specificity by re-expressing TCRs in cell lines such as Jurkats or SKW-3 or fresh primary PBMCs for functional testing [45,46].

In the CD4+ compartment, Tregs are an immunosuppressive subset defined by constitutive expression of FOXP3 that suppress inflammatory responses by a combination of cytokine secretion, including IL-10, TGF-*β*, and inhibitory receptor engagement. In mouse models, Treg depletion by delivering anti-CCR8 monoclonal antibodies re-invigorates tumour-specific effector memory T cells, resulting in profound tumour growth inhibition [47]. Accordingly, several Treg depletion strategies via direct administration of anti-CD25 antibodies [48] or infusion of Treg-depleted autologous T cell products are explored in clinical trials, although with limited success to date. Recent studies have shown that Tregs tend to be the most clonally expanded subset amongst all CD4+ TILs [49,50]. Tregs tend to express a distinct TCR repertoire to conventional CD4+ T cells for the top expanded clones [51,52], suggesting that dominant Treg clones in tumours do not arise from peripheral conversion of highly expanded conventional T cell (Tconv) clones. Multiple reports document the negative prognostic value of Treg infiltration and clonal expansion in multiple tumour types, including RCC [53], colorectal cancer [54], and breast cancer [55].

On the other hand, other CD4+ T cell subsets play crucial roles in effective immune responses, both by providing help to CD8+ T cells and direct cytotoxicity [56]. Notably, immunisation of mice with irradiated squamous cell carcinoma has been shown to induce stem-cell memory-like neoantigen-tetramer positive CD4+ T cells, which provide durable protection against live tumour challenge and re-challenge, via CD40L-dependent help of endogenous CD8+ T cells [57]. Adoptive cellular therapies containing T_H_1 neo-antigen-specific CD4+ T cells have also been shown to induce tumour regression in several case reports [58,59] and a Phase I/II trial [60].

Overall, studying TCR diversity, clonal composition, and repartition in tumours is emerging as an important tool to understand determinants of an effective anti-tumoural response, although there is a critical need to integrate this information with knowledge of the phenotypic states and experimentally confirmed reactivity of T cell subsets.

## 4. TCR Selection Criteria for TCR Based Immunotherapies

Insights gained from profiling the intra-tumoural TCR repertoire not only inform immune dynamics within the tumour microenvironment (TME) but also offer a foundation for therapeutic translation, flipping the paradigm from rational antigen selection approaches to TCR selection strategies. This sets the stage for defining key criteria by which TCRs can be identified, selected, and prioritised for translation as TCR immunotherapies.

Gene transfer of TCRs into autologous T cells derived from peripheral blood offers several practical and therapeutic advantages over the direct use of TILs. Firstly, PBMCs are more readily accessible through minimally invasive procedures, circumventing the need for surgical tumour sampling. Secondly, TILs often exhibit profound functional exhaustion within the TME while expressing TIL-derived TCRs in fitter, non-exhausted peripheral T cells can preserve antigen specificity while restoring functional capacity [61,62,63]. Finally, engineered T cells can be further genetically modified to enhance efficacy, persistence, or safety, to address the important concern that the suppressive TME that initially drove T cell hypofunction or exhaustion would likewise impair engineered T cells expressing these TCRs clonotypes upon re-infusion. Examples of T cell engineering strategies to further overcome TME immune suppression have been extensively reviewed elsewhere [64,65,66].

### 4.1. MHC-I- vs. MHC-II-Restricted TCRs

Traditional cancer immunotherapy approaches have focused on isolating MHC-I-restricted TCRs [9,67] or attempting to raise CD8+ mediated responses with cancer vaccines containing MHC-I-restricted epitopes, although several approaches selecting TCRs restricted to MHC-II neo-antigens have now been pursued [59]. Strategies to target CD4 immunity are notably bolstered by evidence in mice and humans that ICOS+ CD4+ T cells are an important correlate of response to CTLA-4 immune checkpoint blockade [68,69] and longitudinal tracking of CAR-T recipients showing that CD4+ responses, particularly T_H_2 CD4+ responses, underscore functional T cell persistence and durable remission in leukaemia [70]. Furthermore, clinical trials of RNA-based neo-antigen vaccines initially designed to include MHC-I epitopes have been shown to elicit a predominantly CD4+ T cell mediated response in the setting of melanoma and RCC [71,72].

Emerging research is also investigating the potential of engineering unconventional T cell subsets for cancer therapy. This includes the development of MHC-I-restricted CD4+ T cells and MHC-II-restricted CD8+ T cells (Figure 4). CD4+ T cells engineered to express MHC-I-restricted TCRs can directly target tumour cells instead of relying on MHC-II expression, typically only constitutively expressed on antigen-presenting cells (APCs) or induced on endothelial cells following inflammatory stress. Furthermore, MHC-II-restricted CD8+ T cells have been shown to produce polyfunctional cytokine responses to oncoviral antigens [73]. Dual targeting of tumour antigens via both MHC classes will likely improve effectiveness in inducing durable responses and eradicating heterogeneous tumours.

### 4.2. Selecting TCRs by Motif Search and Clonal Expansion

A key challenge of TCR selection for TCR-based immunotherapy is distinguishing bystander TCRs from *bona fide* tumour-reactive TCRs. Of the T cells infiltrating a tumour at a given time, only a fraction may actually express TCRs specific to tumour antigens, while others recognise viral, bacterial, or self epitopes unrelated to the tumour and are therefore useless or even harmful when used for TCR therapy.

A first negative selection step is to exclude TCRs with known, unrelated viral or pathogen reactivities by searching for CDR3 sequence matches in public databases or detecting partial sequence matches via known TCR recognition motifs, although this approach is still limited by the quality of annotations in existing TCR databases. Conversely, in virally driven cancers such as HBV or HPV, identification of TCRs specific to viral epitopes can provide a highly specific way to target cancerous cells. By analysing CDR3 sequences, the TCRGP method has allowed identification of HBV-epitope-specific TCRs and their associated transcriptional state in HBsAg+ HCC patients [74].

A further selection step to prioritise potentially tumour-reactive TCRs is based on clonal expansion as a proxy of antigen-driven proliferation in tumours, leading to elimination of singleton or low-abundance TCR clonotypes and focusing on highly expanded clonotypes [75]. Analysis of the spatial distribution of clonal expansion is typically performed to select clones preferentially expanded in the tumour, although tumour-reactive clones can also be found in TCRs shared between the tumour and the periphery, reflecting the biology of the cancer-immunity cycle [76], whereby T cells primed in the draining lymph node infiltrate the tumour via the circulation. Evidence for the prognostic significance of shared clones is supported by the observation that responders to anti-CD4 monoclonal therapy exhibit a greater expansion of shared CD8+ clones, detected in both blood and tumour compared to non-responders [77].

### 4.3. Prioritising TCRs Based on Gene-Expression Signatures

Beyond simple TCR abundance analysis, several single-cell sequencing workflows such as 10X Chromium and CITE-Seq [78] integrate TCR (VD)J sequencing with either whole RNA transcriptome sequencing or surface marker profiling using oligo-tagged antibodies. Such workflows crucially unlock the ability to integrate TCR clonotype information with phenotypic states of T cells expressing them. Tumour-reactive TCRs are typically enriched in antigen-experienced and exhausted subsets, reflecting prior activation and proliferation to cognate antigen, although fine-grained signatures aim to further disambiguate tumour reactivity from viral or bacterial-induced activation.

Several groups have derived transcriptomic signatures of tumour- or neo-antigen-reactive TCRs, by retro-mapping TCRs confirmed to have tumour reactivity with the RNAseq profile of T cells expressing those clonotypes (Figure 5a). For example, the NeoTCR signature learned across metastatic tumour types identified high expression of the chemokines *CXCL13* and *CXCR6*, effector genes like *GZM-A/B/K*, and *PRF1*, and inhibitory markers like *TIGIT*, *PD1*, and *LAG3*, as well as downregulation of stemness markers including *IL7R* and *KLF2* [44].

Similarly, the PredicTCR classifier identified tumour-reactive TCRs in TIL by learning a gene signature associated with experimentally confirmed reactivity performed by screening TCRs against BT21, a cell line derived from metastatic melanoma [75]. Explainable AI then identified the top 10 transcripts whose abundance was predictive of tumour-reactive T cells including *CXCL13*, *ANXA1*, *IL-7R*, and *TPT1*, although the broad applicability of this approach in diverse cancer types still needs to be explored. Tumour reactivity signatures can then be used to prospectively score and prioritise TCRs in large sequencing datasets, although they importantly require experimental validation of tumour reactivity to take TCRs forward for clinical translation.

### 4.4. Prioritising TCRs Based on Surface Marker Activation Phenotypes

Several functional assays have been developed to experimentally validate TCR cancer reactivity, involving direct culture of TIL purified from tumour or re-expression of TIL-derived TCRs in T cell lines. These are screened against autologous tumour lines for antigen-agnostic approaches, or APCs pulsed with cancer peptides of interest. Such activation-directed workflows, sorting T cells based on markers like CD69 or 4-1BB, can either be used to identify or confirm reactivity of candidate TCRs (Figure 5b,c).

An advantage of identifying molecular signatures of tumour reactivity based on *cell surface* markers is allowing enrichment of viable T cells using FACS, as opposed to RNAseq protocols which require cell lysis. A signature of the integrin CD103+ and the ectonucleotidase CD39+ in CD8+ TILs has been shown to be associated with clones most expanded in the tumour relative to the periphery, expressing high levels of exhaustion markers and a T_RM_ phenotype, and enriched in cells able to kill autologous melanoma tumours [79]. While pipelines of tumour-reactive TIL enrichment can be useful for initial TCR discovery, these have limited applications in directly producing viable T cell products for cell therapy.

## 5. T Cell Engineering

Once one or multiple lead TCRs with tumour reactivity have been identified, extensive pre-clinical screening and optimisations are required to develop a safe and efficacious TCR–T cell product. If the target peptide is known, it is possible to modify the TCR to optimise the affinity of the TCR–pMHC interaction. The potency of a T cell response is critically dependent upon affinity, namely the physical strength of the interaction between the TCR and its pMHC ligand, a quantity denoted by the binding constant K_D_ (K_D_ = k_off_/k_on_) and experimentally measured by surface plasmon resonance (SPR). Importantly, the overall T cell response is dictated not by affinity alone but avidity, which encompasses the cumulative strength of multiple TCR–pMHC interactions, further modulated by co-receptor engagement.

While the affinity of antibodies for their cognate antigen sits in the nanomolar (10^−9^) range, TCRs typically have much lower affinity, in the micromolar (10^−6^) range. An additional hurdle in the context of cancer immunotherapy is that TCRs specific to self peptides typically have even lower affinity than those against non-self epitopes, because of pruning during negative thymic selection. To overcome low endogenous affinity, techniques widely adopted in the monoclonal antibody field using directed evolution of CDRs by phage display have successfully increased TCR affinity up to 10^6^-fold [80], although such high-affinity TCRs are only employed in soluble TCR modalities but are ineffective as TCRs in T cells.

Excessive affinity optimisation is undesirable, as studies including those using model TCRs against the HTLV-1-derived Tax peptide have shown that T cell potency declines at the upper end of the TCR avidity spectrum [42,81], leading to a loss of response to low-antigen-density targets. As T cell activation requires the cumulative engagement of 200 TCRs [82], very strong interactions with APCs prevent serial T cell triggering, leading to a failure of the T cell to reach its activation threshold (Figure 6).

One interesting approach is to turn to Tregs to discover TCRs with high affinity. Tregs are preferentially selected from thymocytes exhibiting high-affinity TCR interactions with self pMHC during positive selection. Hence, TCRs derived from clonally expanded tumour-infiltrating Tregs could be transferred into the patients’ autologous conventional T cells, thereby producing therapeutic T cell products with a higher affinity for tumour-associated self antigens.

### 5.1. Negatively Selecting for Cross-Reactivities Against Healthy Tissue

An essential consideration for pre-clinical assessment of lead TCR safety is assessing its cross-reactivities, in light of the promiscuous nature of TCR binding to pMHC [83] and, most importantly, precedents of severe and even fatal cases of both on-target, off-tumour [84,85] and off-target toxicities [86]. Notable reports include affinity-enhanced TCRs against MART-1 and gp100 causing uveitis and hearing loss due to targeting of epitopes present on healthy melanocytes in the skin, eyes, and ear. Furthermore, a fatal case of cardiac toxicity against a titin-derived peptide due to off-target cross-reactivity by a HLA-A*01-restricted MAGE-A3 TCR reported in 2013 served as a major cautionary tale and has since resulted in major scrutiny into safety assessments of TCRs [86].

Exhaustive cross-reactivity assessment is still highly challenging to perform experimentally and remains an unsolved computational problem. For TCRs with defined specificity, surrogate assays include testing TCRs for recognition against altered peptide ligand (APL) libraries containing substitutions at each residue position of the cognate peptide, often replaced with alanine due to its small size and chemically inert side chain, or algorithmic search for reactivity motifs [87]. Functional screening against cell lines or primary cells can also be performed and forms the basis of the recent clinic approval of Tecelra (afamitresgene autoleucel) against synovial sarcoma, the first TCR-T cell therapy approved in solid cancer [9]. Testing for cross-reactivity against allogeneic cell lines expressing mismatched HLAs can uncover allo-reactivities from therapeutic TCRs and inform clinical contra-indications or exclusion criteria, for example, HLA-A*02:05 allotypes for Tecelra.

In the TCR-centric approach described here, therapeutic TCRs will be identified in TILs of patients and then used to genetically engineer their autologous T cells. Hence, the selected TCRs have already undergone the thymic selection process that results in tolerance to self MHC and to peptides derived from self proteins. As a result, screening for allo-reactivity is not required for these TIL-derived autologous TCRs. However, it is possible that the selected TCRs may recognise tumour-associated self antigens that are also expressed in patients’ healthy tissues outside the thymus. One approach to mitigate this cross-reactivity risk could involve producing autologous pluripotent stem cells (iPSCs) from healthy tissues of patients. The differentiation of the iPSCs into mature cells representing a variety of normal tissues would provide a platform to test TCR-transduced T cells and select against TCRs that recognise non-cancer autologous cells.

### 5.2. Genetic Engineering of TCRs to Manufacture Therapeutic Cell Products

Taking TCRs forward into an adoptive cell therapy product requires a genome engineering strategy to express TCRs in T cells.

The dominant approach for permanent receptor expression, similar to the state of the art in the CAR field, is ex vivo retroviral or lentiviral transduction, which offers durable expression but suffers the limitation of semi-random integration in the genome, potentially leading to insertional oncogenesis, although the safety profile of engineered T cells has been very good to date, with higher risks of secondary malignancy due to radiation or chemotherapy than due to viral integration [88]. In 2017, Eyquem et al. proposed an attractive alternative by achieving CRISPR-Cas9 mediated knock-in (K.I) of the CAR in the endogenous T cell receptor *α* constant (*TRAC*) locus, by providing an AAV template encoding the CAR [89]. In addition to ensuring single integration at a defined genomic site, expression of the CAR becomes subject to endogenous regulation patterns, thereby preventing tonic signalling and improving in vitro cytotoxicity and in vivo persistence.

In the context of TCR-T therapies, the approach of simultaneously knocking out (K.O) the endogenous TCR to replace it with a transgenic TCR construct additionally reduces competition for CD3 for surface expression and the risk of mispairing. The strategy of engineering single or dual *TRAC/TRBC K.O* T cell products is now widely adopted in several in-human trials [90,91]. A crucial limitation of CRISPR/Cas9-mediated K.O that has recently come to light is the risk of genotoxicity due to off-target introduction of double-stranded breaks, producing large-scale chromosomal translocations and aneuploidies [92]. Subsequent genetic engineering strategies for durable expression have innovated by providing non-viral TCR templates for K.O by homology-directed repair (HDR), including dsDNA and ssDNA [93] templates, the latter being less toxic to T cells, or relying on transposons [94]. Transient genetic reprogramming strategies have also been used by electroporating T cells with mRNA encoding TCRs [95,96], reducing risks of genotoxicity and providing a more precise handle on T cell doses, although potentially limited in the durability of T cell protection induced.

## 6. Conclusions

This review highlights the potential of TCR-focused strategies to drive effective anti-tumour immunity by prioritising the understanding of T cell distribution and clonal dynamics within tumours over traditional antigen-centric approaches. We outline methods to identify tumour-reactive TCRs and strategies to further enhance them for therapeutic function. Building on advances from the CAR-T field, additional T cell engineering steps, including signalling amplification, cytokine arming, inhibitory receptor knock-outs, and metabolic reprogramming, all offer further means to enhance TCR-T cell fitness and function. With the advance of robust computational tools and assays to predict and confirm TCR reactivity, TCR-based approaches offer a compelling foundation for the development of more precise and durable cancer immunotherapy.

## Figures and Tables

**Figure 1 cells-14-01223-f001:**
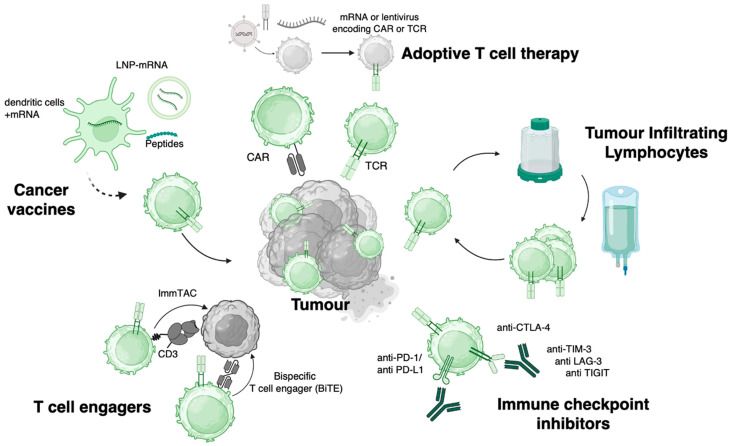
T cell engaging modalities in cancer immunotherapy. This graphic represents immune-engaging modalities directly or indirectly converging onto the activation of anti-tumoural T cells. In adoptive T cell therapy, patient-derived T cells are transduced or electroporated to express Chimeric Antigen Receptors (CARs) or T cell receptors (TCRs) specific to tumour proteins, then adoptively transferred to a patient. In tumour-infiltrating lymphocyte (TIL) therapy, TILs collected from surgical resections are expanded ex vivo, then re-infused into the patient. Immune checkpoint inhibition therapies deliver monoclonal antibodies blocking inhibitory receptors including CTLA-4, PD-1/PD-L1, TIM-3, LAG-3, and TIGIT, thereby relieving functional inhibition of endogenous T cells. Similarly, T cell engagers are bispecific molecules, containing a CD3 binding moiety and a cancer target binding moiety, either derived from scFv for traditional bispecifics or from a TCR for ImmTACs, resulting in local recruitment and activation of endogenous T cells. Lastly, therapeutic cancer vaccines in the form of mRNA, peptide, or dendritic cells loaded with cancer antigens all aim to prime or boost endogenous tumour specific T cells.

**Figure 2 cells-14-01223-f002:**
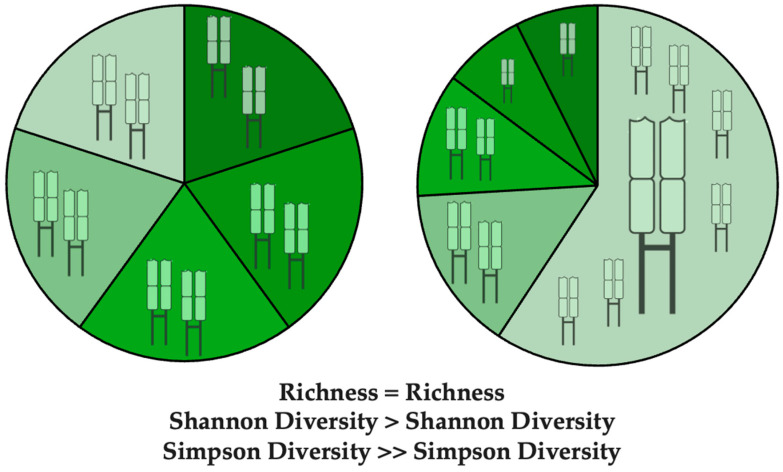
Schematic of richness and evenness diversity metrics in TCR repertoires. The two illustrated TCR repertoires have the same richness, each containing 5 unique clonotypes, represented by unique colours in sectors of the pie charts. Despite equal richness, the left repertoire (**left**) has higher evenness, in which all clonotypes make up the same proportion of the repertoire (20%), while the right repertoire (**right**) has a more uneven repertoire, which is dominated by a single expanded clonotype, and made up of additional clonotypes with uneven frequencies. Measures such as Simpson and Shannon diversity capture the intuitive notion that the more uneven repertoire is less diverse.

**Figure 3 cells-14-01223-f003:**
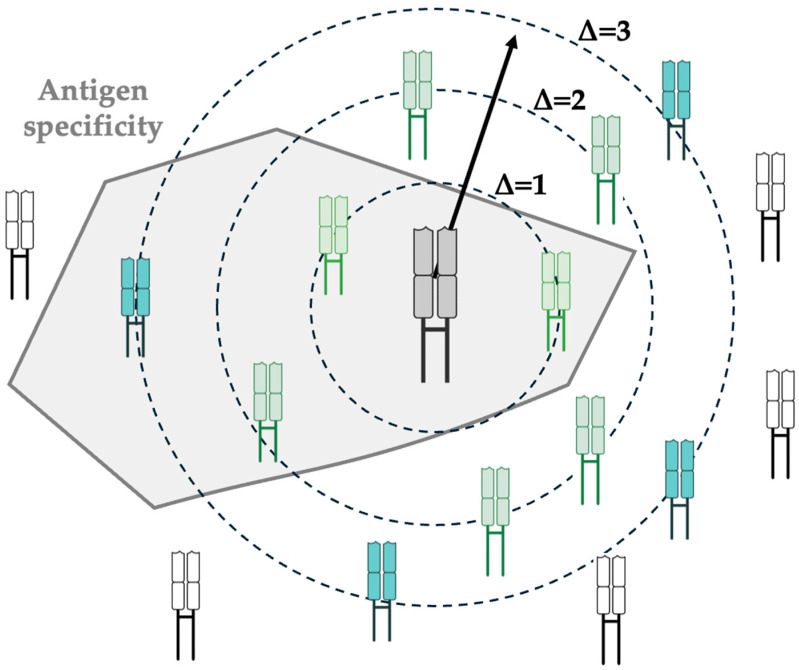
Schematic representation of a TCR meta-clonotype. A TCR meta-clonotype is an ensemble of TCRs with high sequence similarity, or sharing sequence and structural motifs, which are inferred to share antigen specificity. A “centroid” TCR is illustrated, and concentric rings of TCR sequences differing from the centroid TCR by various edit distances: (∆ = 1, 2, 3 substitutions) are illustrated here, and colour-coded according to edit distances. The edit distance may be calculated by comparing the TCR*α* sequence, the TCR*β* sequence, or both, and commonly uses the Levenshtein distance, a metric quantifying the difference between string sequences. TCRs with experimentally determined antigen specificity, represented by a grey boundary, may encompass TCRs with varying edit distances, and this does not necessarily correlate with the number of substitutions.

**Figure 4 cells-14-01223-f004:**
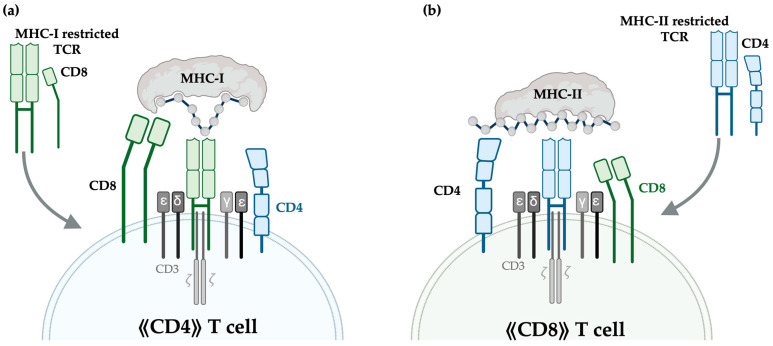
Schematic of unconventional MHC-I and MHC-II-restricted T cell engineering. (**a**) Schematic representation of providing an MHC-I-restricted TCR, alongside a CD8 co-receptor, in a CD4+ helper T cell endogenously expressing CD4. Provision of the MHC-I-restricted TCRs allows this T cell to recognise peptide–MHC-I complexes, constitutively expressed on nucleated cells, while preserving helper or memory functions. The CD8 co-receptor stabilises the interaction with peptide–MHC-I complexes. (**b**) Provision of an MHC-II-restricted TCRs, alongside a CD4 co-receptor, in a CD8+ T cell endogenously expressing CD8. Provision of the MHC-II-restricted TCRs allows this T cell to recognise peptide–MHC-II complexes, while simultaneously preserving cytotoxic functions. CD3εδγ*ζ* moieties required for TCR signal transduction are illustrated.

**Figure 5 cells-14-01223-f005:**
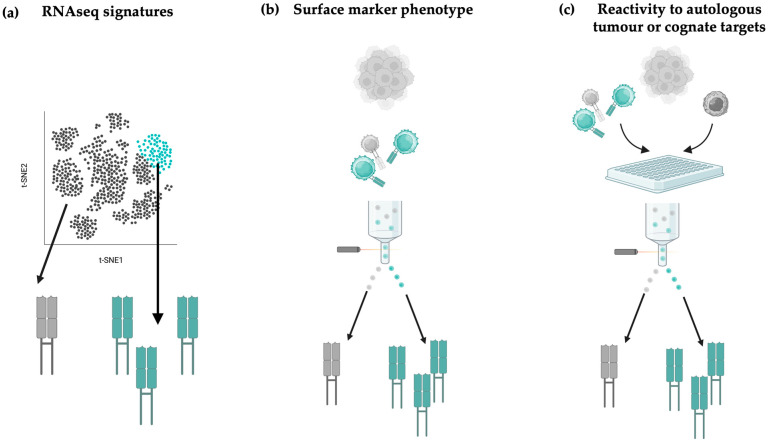
Tumour-reactive TCR identification methods. (**a**) RNAseq-based identification methods. TILs or PBMCs sequenced with dual V(D)J and gene-expression (GEX) profiling can be scored on gene-expression signatures of T cells experimentally confirmed to display tumour reactivity. TCRs belonging to clusters with inferred tumour-reactive phenotypes can then be taken forward for screening. (**b**) Flow-cytometry identification methods. Primary TIL or PBMC isolated from patients can be stained for surface markers of putative tumour-reactive signatures, such as CD103, CD39, or CD40L for fluorescence-activated cell sorting (FACS) and TCR sequencing, or cultured for further assays. (**c**) Primary TILs, PBMCs, or Jurkat lines transduced with TCR pools can be initially co-cultured with autologous cancer targets or cell lines loaded with cancer antigens, followed by staining for activation markers including CD69 or 4-1BB, followed by TCR sequencing, or cultured for further assays. In all diagrams, grey TCRs schematically represent bystander TCRs and green TCRs schematically represent tumour-reactive TCRs.

**Figure 6 cells-14-01223-f006:**
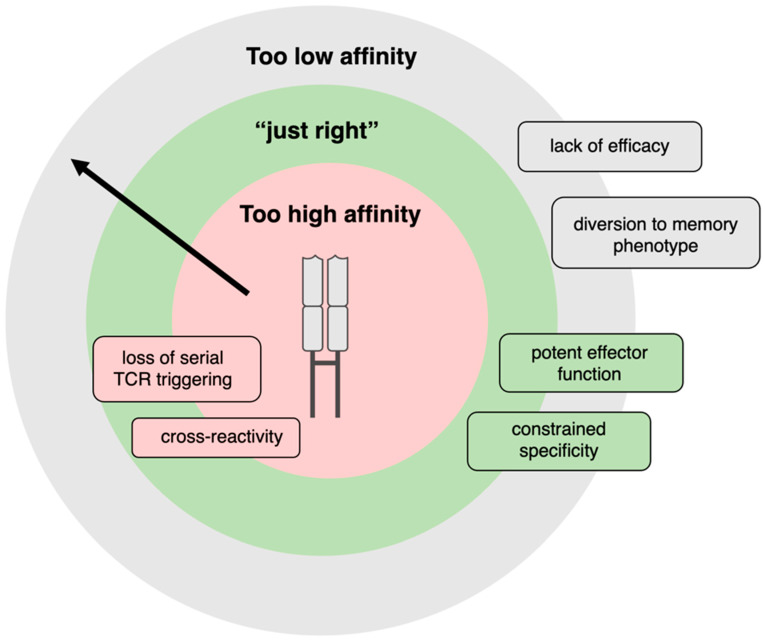
The Goldilocks’ zone of TCR avidity. Schematic illustrating the avidity of TCRs for target pMHC antigen as concentric circles, with high avidity rings closer to the central TCR and lower avidity furthest away. The name Goldilocks’ zone is taken as an analogy to represent the zone of TCR avidity that is neither “too high” nor “ too low”, but well suited to elicit effective T cell activation. Very-high-avidity interactions lead to the gain of undesirable and toxic cross-reactivities and make T cells lose responses to low-antigen density targets. Very-low-avidity interactions are ineffective at signalling altogether, or induce T cells to differentiate into memory rather than effector phenotypes. A narrow avidity window is optimal to induce potent T cell activation and maintain peptide specificity. The functional avidity window is not a TCR-intrinsic feature and depends on additional factors such as T cell activation state and inflammatory cues in the milieu.

## Data Availability

No new data were created or analyzed in this study.

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
