# Peer review of "Exploiting TCR Repertoire Analysis to Select Therapeutic TCRs for Cancer Immunotherapy"

_cells, 2025, doi:10.3390/cells14151223_

Round 1
Reviewer 1 Report
Comments and Suggestions for Authors
In this review, Ursule Demael and colleagues discuss the conceptual framework and technical challenges (and their potential solutions) underlying the novel T-cell receptor (TCR) based T cell therapeutic avenue for cancer treatment. The authors elegantly provide the context and rationale for screening, identification, prioritization, and validation of therapeutic TCRs from patient tumor-infiltrating lymphocytes, and further highlight potential pitfalls and possible technological solutions to overcome them. This review is well organized, clear, concise, and touches on important themes relevant to developing personalized cell therapies in solid tumors, where state-of-the-art CAR T cell therapies exhibit minimal (in some cases modest) efficacy.
Below are some pointers that may help enhance the quality and reliability of the review.
- The introduction section (1) is empty. It may be useful to relocate some of the basic and preliminary discussion on the T cell and TCR repertoire in the tumor from section (2) 'Main Text' to the introduction and lay the foundation of the central theme of the article early on.
- Throughout the text, the authors emphasized that T cells (in turn, TCR clonotypes) responding to tumor antigens (in TIL) have mostly been found to be in an exhausted state. This implies that such clonotypes, despite possessing specificity and reactivity to tumor antigens, might not withstand the suppressive tumor microenvironment (mostly driven by tumor cells themselves but also dynamically being shaped by other immunosuppressive cells, such as cancer-associated fibroblasts and suppressive myeloid populations, etc., in the TME). It would be highly counterintuitive to imagine placing such clonotypes (for example, after TCR engineering the autologous T cells) into the same suppressive environment, where TME is supposed to be (or already) capable of (perhaps equipped) with cellular and molecular arsenals to push such therapeutic TCR clonotypes into hypofunctional, exhaustive, or even exclusion from the active center of tumor. It would be interesting if the authors include a discussion on potential remedies to overcome some of these key issues. The key question here is what would make it highly likely to have better therapeutic outcomes? Whether we need to improve the quantity and/or quality of therapeutic TCR clonotypes, or whether we should also think about simultaneously enhancing their resilience in a suppressive environment?
- It would also be useful (if the authors feel that this might fit in the scope of this review) if the authors discuss the possibility of targeting a pool of multiple therapeutic TCRs as a therapeutic means to address escape from immune surveillance, with mechanisms such as immuno-editing or antigen sequestration, etc.
Author Response
Thank you for your comments. We have answered in the attached document

Reviewer 2 Report
Comments and Suggestions for Authors
The aim of the paper is to review the potential of strategies starting from the analysis of TCR repertoire of individual patients to develop new cancer immunotherapeutic approaches based on TCR engineering.
There is no doubt that immunology-based strategies such as checkpoint inhibition and CAR T cell production met with important successes and currently raise much interest. There is admittedly less focus on many approaches discussed in this review, and this provides interesting material that is less well known.
However, while this review is undoubtedly useful, it is sometimes difficult to read and it is not easy to derive a clear takehome lesson. The authors give a great amount of information, sometimes well known to most immunologists and sometimes more specialized, and it is often difficult to follow their line of thought. Another point is that it is often difficult to discriminate between well supported strategies and more speculative hypotheses. Below are a number of specific points that illustrate this opinion.
Specific points:
- The manuscript should be carefully reread:
The section entitled "1.Introduction" is empty.
A number of bibliographical references should be corrected. There is no publication year in refs. 4 and 5, the title of ref.9 should be corrected.
- Methods sketched in Figure 1 are very diverse, and the link with TCR specificity is not always obvious.
For example, the success of CARs in haemotological malignancies rather that solid tumors may be related to environmental factors, not specificity (Joy24, Cancer Res. 84:2432-2449). The consequences of the structural differences between CARs and TCRs should also be mentioned (this point is illustrated by a number of reports, such as Harris18 J. Immunol. 200:1088-1100, Davenport 18 Proc. Natl. Acad. Sci. USA 115:E2068-E2076, Burton23 Pnas 120:e2216352120).
- The authors emphasize the capacity of CD8+ Tcells to recognize intracellular antigens (lines 95 and 118), and this is certainly relevant to their point. Otherwise, the purpose of the section entitled "TCRs govern T cell function" (line 102) is not clear.
- The point raised in lines 140-141 is interesting, but is the major bottleneck the need to identify the target of "orphan" TCRs or only the interest of identifying TCRs with anti-cancer potential whatever their specificity ?
- Lines 154-156 : why is the analysis of TCR diversity an important tool ? the rationale is not easy to understand.
- Line 176: what is the sampling depth ? do the authors mean that if your analyse a higher number of TCRs it is likely that you will find a higher number of different sequences. Perhaps the authors might provide a mathematical formula, if it is understandable ?
- Line 186 : even if there is no "robust consensus", the authors should provide several references to support the hypothesis that diversity and evenness deserve being considered. An obvious question (line 212) is whether the number of clones with tumour reactivity is more significant that the number of cells with tumour reactivity.
- Line 259 : while it seems reasonable to group TCRs with similar CDR3 sequences or shared motifs, what are "comparable biochemical features" ?
- Line 269: what is Levenshtein distance ?
- line 272 - 287 : the authors might discuss more thoroughly their prediction (line 283) that affinity and/or avidity are important parameters (the paper by Beppler et al., 2023 (. Cell Biol. 222:e202205118) exemplifies this point, but this is supported by many other reports)
- Line 298 : "singleton" should be explained
- Line 299: "dysfunctional ?"
- The authors mention Tregs (line 311) before describing some key properties of this subset (319). While this is may be a matter of taste, I think that it might be preferable to reverse the presentation order.
- Line 343 : does TCR repertoire profiling consist of determining diversity and evenness, as explained above, or does this include aims at quantifying binding parameters and functional properties ? This is much more demanding and experimental approaches must be discussed.
- Lines 349-356: this point is reasonable. Is this supported by recent experimental studies ?
- Line 402 : This seems reasonable, but see Caruso15 (Cancer Res. 75:3505-3519) : a CAR recognizing a "normal" autoantigen might be useful.
- Line 426 : it is not easy to understand how T cell phenotypic state might be derived from sequence studies.
- Line 443 more information is needed to support the hypothesis that experiments described in ref. 62 are generalizable enough to define "tumour reactivity signatures".
- Line 505 : what is Goldilock's zone of TCR avidity ?
Author Response
Thank you for many useful comments. Answered in the attached document

Round 2
Reviewer 2 Report
Comments and Suggestions for Authors
This review may bring the attention of a number of readers on some points that are not familiar with. Thus, publication is recommended.